# Involvement of Microglia in the Pathophysiology of Intracranial Aneurysms and Vascular Malformations—A Short Overview

**DOI:** 10.3390/ijms22116141

**Published:** 2021-06-07

**Authors:** Teodora Larisa Timis, Ioan Alexandru Florian, Sergiu Susman, Ioan Stefan Florian

**Affiliations:** 1Department of Physiology, Iuliu Hatieganu University of Medicine and Pharmacy, 400006 Cluj-Napoca, Romania; 2Clinic of Neurosurgery, Iuliu Hatieganu University of Medicine and Pharmacy, Cluj County Emergency Clinical Hospital, 400012 Cluj-Napoca, Romania; stefanfloriannch@gmail.com; 3Department of Histology, Iuliu Hatieganu, University of Medicine and Pharmacy, 400012 Cluj-Napoca, Romania; serman_s@yahoo.com

**Keywords:** microglia, brain arteriovenous malformation, cerebral cavernous malformation, subarachnoid hemorrhage, vasospasm

## Abstract

Aneurysms and vascular malformations of the brain represent an important source of intracranial hemorrhage and subsequent mortality and morbidity. We are only beginning to discern the involvement of microglia, the resident immune cell of the central nervous system, in these pathologies and their outcomes. Recent evidence suggests that activated proinflammatory microglia are implicated in the expansion of brain injury following subarachnoid hemorrhage (SAH) in both the acute and chronic phases, being also a main actor in vasospasm, considerably the most severe complication of SAH. On the other hand, anti-inflammatory microglia may be involved in the resolution of cerebral injury and hemorrhage. These immune cells have also been observed in high numbers in brain arteriovenous malformations (bAVM) and cerebral cavernomas (CCM), although their roles in these lesions are currently incompletely ascertained. The following review aims to shed a light on the most significant findings related to microglia and their roles in intracranial aneurysms and vascular malformations, as well as possibly establish the course for future research.

## 1. Background

Intracranial aneurysms are pathological dilations of arteries and represent the point of minimal resistance of their parent vessel, as well as the greatest source of non-traumatic subarachnoid hemorrhage (SAH). Aneurysms possess an estimated 3% prevalence in the general population and, to this day, SAH is associated with a high mortality and morbidity, being capable of inflicting long-term and occasionally irreversible neurological damage [1,2,3,4,5]. Hemodynamic alterations and vascular wall degeneration have both been incriminated in the etiology of aneurysms, as hemodynamic and histological studies show their preferential placement where the blood flow is more turbulent, as well as the absence of the tunica media in their walls [6,7,8]. However, more and more evidence supports an important contribution of inflammatory processes leading to their occurrence [9,10].

Brain arteriovenous malformations (bAVM) and cerebral cavernous malformations (CCM) are predominantly congenital lesions of the intracranial vasculature, with the first possessing high blood flow, and the latter low to minimal internal flow. Both possess a predisposition for rupture and debilitating cerebral hemorrhage based on their location, bAVMs being especially prone to this complication. Because they are comprised of serpiginous and entangled vessels shunting oxygenated blood directly into the venous stream, bAVMs represent a therapeutic challenge from all perspectives, surgical, endovascular or radiosurgical [3,11,12,13]. CCMs, on the other hand, are theoretically much easier to resect since they possess extremely thin vascular feeders, yet this means that endovascular procedures are not an option [3,14,15]. Many breakthroughs have been made in understanding these two distinct pathologies, yet their pathophysiology and etiology still largely remain a mystery.

Microglial cells, or microglia, represent resident immune cells of the cerebral parenchyma, and, alongside perivascular macrophages, belong to the innate immune system [16,17,18]. These are the only glia that have a myeloid origin, developing from the extraembryonic yolk sac exclusively during a surprisingly restricted interval [19,20]. Other sources of origin have also been proposed, such as the neuroectodermal or mesodermal precursor cells [21]. They are crucial in maintaining the homeostasis of the central nervous system (CNS) microenvironment, possessing the capacity to shift their function toward either the pro- (M1) or anti-inflammatory (M2) phenotype to protect the brain accordingly. They are also able to perceive alterations in the external environment and subsequently adapt their role within the internal microenvironment [22]. However, the drawback is that a disproportionate activation of the proinflammatory microglia reacting to primary neuronal or axonal degeneration and other processes associated with systemic inflammation can initiate or sustain chronic inflammation [16]. As such, this phenotype may be correlated with the pathogenesis of certain inflammatory neurological disorders like Alzheimer’s disease, amyotrophic lateral sclerosis, epilepsy, stroke, and brain tumors. While much has already been discovered regarding the implications of microglia in aneurysmal SAH, there is a current scarceness of data related to the roles they play in the development of intracranial vascular malformations.

This short narrative review aims to provide an illustrative summary on the involvement of microglia in the pathophysiology of intracranial aneurysms, SAH, and cerebral vascular malformations as well as propose future directions of research and treatment for these conditions. We performed a thorough research of the electronic databases of Medline (PubMed, PubMed Central), Cochrane Library, and Google Scholar by using permutations of the terms and idioms: ’aneurysm’, ‘subarachnoid hemorrhage’, ‘SAH’, ‘arteriovenous malformation’, ‘bAVM’, ‘cavernous malformation’, ‘cavernoma’, and ’CCM’ in one field, and ’microglia’ on the other. Next, we employed a cross-reference search to include similar or associated studies on these subjects. We have decided to include both clinical and paraclinical investigations on adult human patients and their respective pathological tissue samples, as well as experimental studies performed in vitro on mice and rats, and in vivo. The paragraphs may provide a meaningful insight on the lesser-known aspects of brain inflammation and immunologic response correlated with these pathologies.

## 2. Vasculogenesis and Angiogenesis as Viewed by Microglia

As to understand and possibly provide new directions for research, we think it necessary to highlight current evidence regarding the contribution of microglia in the processes of vasculogenesis and angiogenesis. Vasculogenesis conveys the de novo development of vessels from differentiating and migrating angioblasts [23]. Angiogenesis, on the other hand, represents the formation of brand-new vessels from already existing ones, starting as early as day 9.5 of embryonal development, and it is the exclusive process through which the cerebral vasculature is formed [24,25]. It is in this phase when the first microglial progenitors are observed closely associated with developing blood vessels, suggesting that they indeed possess a valuable role in cerebral vessel formation. It has been shown that microglial depletion results in a sparser vascular network [26,27]. Duduki et al. found that the lack of intracellular adapter Kindlin3, which within the CNS is exclusively expressed by microglia, leads to increased microglial contractility, dysregulation of the ERK pathway, overexpression of transforming growth factor-beta 1 (TGFβ1), and malformed vasculature within the retina [28]. This phenomenon could be remedied by either microglial depletion or by microglia-specific knockout of TGFβ1 in mouse models. Nonetheless, the mechanisms through which microglia shape the cerebral vasculature should be further investigated.

## 3. Birth by Fire—Are Microglia Responsible for Intracranial Aneurysm Development?

While M1 proinflammatory macrophages have been incriminated as a requirement for the formation of aneurysms in a CXCL1 dependent neutrophil inflammation model in mice [29], it is unclear whether microglia themselves share this characteristic. Hasan et al. found that both ruptured and unruptured aneurysm walls had positive immunohistochemical staining for both M1 (anti-Human Leukocyte Antigen—DR isotype (HLA-DR) antibodies) and M2 (anti-CD 163 antibodies) macrophages [30]. Moreover, unruptured aneurysms presented a relatively equal staining for both phenotypes, whereas the ruptured ones had a more abundant staining for M1 cells. Indeed, this finding may not apply universally, as Yamashiro et al. discovered that the difference between M1 and M2 macrophage populations was not homogeneous within their patients, in some cases these two phenotypes presenting reversed or relatively equal numbers in the aneurysmal walls [31]. This might imply that the allegedly benign M2 phenotype tasked with repair actually plays a currently unknown role in vascular wall degeneration and aneurysm formation.

Regarding the molecular aspects of macrophage polarization, it has already been shown that toll-like receptor 4 (TLR4) compels macrophages skewedly toward the M1 phenotype in intracranial aneurysms [32,33,34]. The signaling cascade via the myeloid differentiation primary response gene 88 (MyD88) leads to the activation of inhibitor kappa B kinase *β* (IKK*β*), which in turn triggers the phosphorylation and successive degradation of inhibitor kappa B (I*κ*B). This then translates into the translocation of nuclear factor kappa-light-chain-enhancer of activated B cells (NF-*κ*B) into the macrophage nucleus, thereby upregulating the expression of several genes involved in inflammation and consequential tissue damage [35]. Hemodynamic stress is also crucial in the formation of aneurysms by activating prostaglandin E2 (PGE2) and prostaglandin E receptor 2 (EP2) signaling and amplifying chronic inflammation through NF-*κ*B [36].

Alternatively, the M2 phenotype is stimulated by other factors, such as peroxisome proliferator-activated receptor γ (PPAR*γ*), or pioglitazone, an agonist of PPAR*γ*, which resulted in a protective effect against aneurysms in mice [37,38]. In addition, extracellular signal-regulated kinase 5 (ERK5) upregulation increased the M2 population relative to the M phenotype ratio by restraining the NF-*κ*B pathway in intracranial aneurysms [39]. In the same study, anagliptin, a dipeptidyl peptidase-4 (DPP-4) inhibitor and a known anti-atherogenic agent [40,41], inhibited the accumulation of macrophages in the walls of aneurysms and also diminished the size of aneurysms in mice [39]. Another DPP-4 inhibitor, teneligliptin, was demonstrated to hinder macrophage infiltration and the formation of abdominal aortic aneurysms in mice [42], although its effect on intracranial aneurysms has, to the best of our knowledge, not been examined yet. Despite the fact that the roles played by macrophages in aneurysm occurrence have been rigorously researched so far, the involvement of microglia *per se* has not yet been revealed. It stands to reason that the mechanisms implicated in this process might be shared between the two types of immune cells.

## 4. Microglia in the Early Stage of SAH—The Arrival of the Storm

### 4.1. Cerebral Inflammation after SAH

The M1 phenotype of microglia is associated with the increase and expansion of cerebral inflammation following SAH, leading to secondary brain injury [9,10,43,44]. Microglia activation is especially prevalent within the regions in proximity to the hematoma or the SAH inundated basal cisterns and as early as 24 h after the initial rupture. This then furthers processes of neuronal apoptosis and necrosis, resulting in a subsequent discharge of inflammatory molecules (alarmins) from the nuclei, like high-mobility group box 1 (HMGB1), which in turn sustains the activation of additional microglia [45]. This self-sustaining process then leads to the recruitment of perivascular macrophages, increased blood–brain barrier permeability, as well as heightened neuronal loss [44,46]. Despite emerging evidence that microglia can be phagocytic, there is no indication that they execute this task in the early stage after SAH. Erythropoietin (EPO) has been shown to ameliorate brain injury in the acute stage following SAH in mice by promoting the polarization of microglia toward the anti-inflammatory M2 phenotype, at least in part via the EPO Receptor/Janus Kinase 2/Signal transducer and activator of transcription 3 (EPOR/JAK2/STAT3) signaling pathway [46]. This would suggest a highly beneficial outcome in the treatment of patients harboring ruptured aneurysms and SAH with EPO, especially in the case of early brain injury, although the pathways and mechanisms involved have yet to be definitively demonstrated.

### 4.2. Autophagy and Neuronal Cell Death

Mammalian target of rapamycin (mTOR), a negative regulator of cellular autophagy, is increased within the brain parenchyma after the occurrence of SAH [47]. In a recent in vitro study of SAH induced in rats, rapamycin or AZD8055 significantly diminished the phosphorylation levels of mTOR, while also stimulating the microglial polarization in the direction of the M2 phenotype and alleviating early brain injury [48]. Vascular endothelial (VE)-cadherin, which is an adhesion protein, is cleaved during inflammation into the soluble fragments (sVE-cadherin) and present in high amounts in the cerebrospinal fluid (CSF) of patients with SAH [49]. The high levels of sVE-cadherin were significantly correlated with increased clinical severity, but not with vasospasm. Furthermore, mouse brains injected with sVE-cadherin brain led to an upsurge of proinflammatory activated microglia adjacent to the injection site.

In rat models with SAH, Zhen-nan et al. ascertain that the activated microglia displayed an obvious increase of Leukotriene B4 receptor 1 (BLT1) expression [50]. BLT1 was also present in neurons and endothelial cells, whereas astrocytes did not exhibit this receptor. It has already been established that BLT1 is expressed on neutrophils, stimulating their chemotaxis and adhesion in reaction to leukotriene B4 in the inflammatory site within the CNS [51]. Whether this denotes an important interaction between microglia and neutrophils in the pathophysiology of SAH should be corroborated in future research.

### 4.3. HMGB1: A Harbinger of Vasospasm?

The cytokine HMGB1 level was amplified in the animal brain after SAH, more specifically in the activated proinflammatory microglia [52]. The elevation of HMGB1 in CSF is associated with a poorer outcome in SAH [53,54,55] and might even stimulate vasospasm [56,57]. Increased levels of HMGB1 in the CSF of these patients can be attributed to either release from M1 microglia or passive diffusion from within the nuclei of necrotic or apoptotic CNS cells [52]. Additionally, HMGB1 has been incriminated in the phenotype switching of vascular smooth muscle cells (VSMCs) occurring in vasospasm after SAH [58]. Antibodies against this HMGB1 not only limited this process, but also reversed vasoconstriction, attenuated vascular remodeling, and limited microglial activation and brain edema. Nevertheless, this cytokine and its effects in SAH represent an intriguing subject to be decidedly validated in subsequent investigations.

A summary of the mechanisms and pathways that have been described in the early stage after aneurysmal SAH can be viewed in Figure 1. After the initial acute stage, the histological alterations noticed in microglial cells denote a dynamism between the activation within 24 h after rupture, followed by a reversion near to baseline at 3 days, and ultimately a reactivation at 14 days [44]. It is after the 3-day mark that vasospasm usually occurs in patients with SAH from ruptured aneurysms, and microglia have been incriminated as playing a crucial role in this complication.

## 5. Microglia and Vasospasm—Rubbing Salt on the Wound

### 5.1. For Whom the Bell Tolls—TLR4 Upregulation Triggers Vasospasm

Vasospasm one of the deadliest complications following SAH, representing an anarchic constriction of the cerebral vasculature that could lead to severe ischemic stroke [59,60,61]. This usually occurs in the subacute stage, from the third day after rupture onward, though rarely after day 14. Microglial cells have displayed a tendency to increase in population over time, especially at the time of vasospasm, 6 days after rupture [59]. According to Hanafy, microglia and TLR4 expression were seemingly equally essential and sufficient to trigger vasospasm within both the early and late stages of SAH in their mice models [62]. It has been demonstrated that TLR4 is upregulated in the brain both in human patients and mice models with SAH [34,62]. Moreover, peroxiredoxin 2, the second most abundant protein within the cerebrospinal fluid (CSF) following traumatic brain injury and SAH, reportedly activates microglia into the M1 type through the TLR4/MyD88/NF-*κ*B pathway [63]. Following SAH, MyD88 can be overexpressed in microglia and astrocytes [64], and the inhibition of NF-*κ*B by CCM3 upregulation may alleviate neurological damage [65]. On the other hand, curcumin, a herbal extract from the roots of Curcuma longa, apparently mitigates inflammation by inhibiting the TLR4 axis, thereby polarizing microglia towards the M2 phenotype [66]. In a similar study, hydrogen sulfide reduced neuroinflammation by inhibiting TLR4/NF-*κ*B signaling in microglia, although its effects on vasospasm were not investigated [67]. Apigenin might suppress TLR4 activation, thereby restricting microglia-mediated apoptosis and inflammation while protecting against BBB disruption [68]. The long non-coding RNA (lncRNA) fantom3_F730004F19 has also been identified as a potential target of therapy, as its knockdown via lentivirus treatment in mice led to the downregulation of CD14 and TLR4 proteins in microglia and consequent inflammation [69]. CSF peroxiredoxin 2 will likely be used as a biomarker to evaluate the severity of vasospasm, whereas curcumin, hydrogen sulfide, or apigenin may become future agents to be used in preventing or alleviating this occurrence.

### 5.2. Damage Control—6-MP and Microglial HO-1

Chang et al. showed that 6-Mercaptopurine (6-MP) moderates microglia chemotaxis and inflammation in SAH rat models in a dose-dependent fashion and separate of nitric oxide synthetase [70], and also attenuates cellular adhesion molecules [71]. Additionally, by lessening endothelin-1 level, 6-MP seemingly exerts an inhibitory effect on vasospasm [70]. Through its combined outcomes, 6-MP may exemplify a worthwhile research subject on both cellular and molecular standpoints in SAH and other brain injuries.

Microglial heme oxygenase isoform 1 (HO-1) and the inhalation of carbon monoxide (CO) are important for the efficient elimination of blood and heme following SAH, thus mitigating neuronal apoptosis and necrosis, vasospasm, and impaired cognitive function [72]. CO can regulate microglial CD36 surface expression to influence erythrophagocytosis, although this phenomenon was reduced in HO-1 deficient mice [73]. In the same research, mice with CD36 knockout also exhibited impaired erythrophagocytosis and exacerbated vasospasm. Conversely, a dual in vivo and in vitro experiment on mouse models with SAH showed that, while intraventricular deferoxamine did limit neurological damage, improve cognitive outcome, and increase HO-1 expression in microglia, it had no effect on vasospasm [74]. It remains to be validated whether HO-1 from microglia plays a role in influencing the severity of vasospasm in humans as well and if it represents a suitable therapeutic target for this dreadful complication. Figure 2 displays the mechanisms and pathways of vasospasm in which microglia are known to be engaged.

## 6. Microglia in the Delayed Stage of SAH—Far from Over

As mentioned earlier, the activation of microglia decreases by day 3, to steadily bounce back and peak at 14 days after the onset of SAH, only to subside near day 28 [44,75,76]. It is within this timeframe that microglia exert their proinflammatory effects on the parenchyma, leading to long-term neurological damage. Between days 14 to 28, microglial cells intensify their expression of interleukin-6 (IL-6), TLR4, as well as tumor necrosis factor-alpha (TNFα) [62,75] thereby increasing the blood-brain barrier permeability and leading to the recruitment of additional peripheral immune cells [77,78]. In mice with knockout for intercellular adhesion molecule-1 (ICAM-1) and P-selectin glycoprotein ligand-1 (PSGL-1), Atangana et al. demonstrated a marked reduction in neutrophil-endothelial interaction within the initial 7 days after experimental SAH, leading to considerably attenuated accumulation of microglia and neuronal cell death [79]. These findings support the longer duration of the chemoattractant and neurodegenerative effect of microglia following SAH, meaning that even a neurologically intact patient within this period is not entirely safe from delayed consequences. There is a growing body of data advocating that microglia are implicated in embryonal angiogenesis, as well as interacting with the vasculature in several tumoral, ischemic, and neurodegenerative pathologies [80]. Despite the fact that the delayed and chronic events after SAH are well-documented, the involvement of microglia in these events continues to be elusive, its pathophysiological mechanisms and pathways requiring additional investigation.

## 7. The Future of Microglia in SAH

Ultimately, it should be stated that because of the inherent difficulty to accurately discriminate between the immune cells of the CNS, it can be questioned whether the alterations accredited to microglia after SAH do not actually stem from the other cells [44]. As there is a paucity of commonly recognized and reliable markers, single-cell RNA sequencing has helped with the identification of differentially expressed genes (DEG) that are distinctive to microglial cells, namely *P2ry12*, *Tmem119, Slc2a5*, and *Fcrls* [81]. A protocol allowing the concomitant purification of viable single-cell suspensions of all key CNS-resident cell types from adult mouse brains, healthy or otherwise, has recently been published [82]. This could have widespread preclinical and clinical implications, including a better understanding of numerous pathologies of the CNS and more specific ways to treat them. Microglia-targeted therapies may also become a valuable treatment method, wherein either inflammatory preconditioning with lipopolysaccharides or pharmacological deactivation with colony-stimulating factor-1 (CSF-1) receptor-antagonist Perxidartinib (PLX3397) can alleviate neuronal death after SAH [9]. Additional studies utilizing the aforementioned markers and protocol would indubitably enhance our knowledge regarding the implications of microglia and their implied neuroinflammatory processes in SAH pathophysiology. Figure 3 collectively summarizes the currently known mechanisms wherein microglia are involved in the pathophysiology and development of intracranial aneurysms, aneurysmal SAH, and vasospasm.

## 8. Microglia in Arteriovenous Malformations and Cavernomas—An Ongoing Enigma

### 8.1. AVMs—The Evidence on Hand

The accumulation of microglia and macrophages has been documented in bAVMs both in humans and animal models, regardless of ruptured or unruptured characteristics [83]. The majority of microglia present in these malformations is active, possibly causing persistent inflammation, which then stimulates anomalous angiogenesis and lesion growth [84]. According to Chen et al., bAVMs exhibited a higher amount of the macrophage/microglia marker CD68 than epilepsy controls [85]. These cells were distributed throughout the bAVM tissue samples, such as the vascular wall and adjacent parenchyma, concurrent with the hypothesis that inflammation plays an as of yet undisclosed role in the pathophysiology of these lesions. In a mouse brain model with focal activin receptor-like kinase 1 (Alk-1)-deficiency, both microglia and bone marrow-derived macrophages were present within bAVMs [86]. However, these could have been both active and dormant resident microglia, as they share a common label, green fluorescent protein (GFP). Moreover, the number of microglia/macrophages is higher in the Alk-1-deficient mouse brain following VEGF stimulation than in those without this deficiency or lacking VEGF treatment [87]. The integrity of bAVM vessels in these brains was compromised, as corroborated by the boosted deposits of fibrin and iron, the microscopic pockets of extravasated red blood cells around the vessels, and the macrophage/microglia infiltration within the vessel wall and in the neighboring parenchyma. It has been also shown that Alk-1 deletion and angiogenic stimulation with VEGF can trigger the formation of de novo bAVMs in mice [88], yet the involvement of microglia in this process in humans is uncertain. The administration of colony-stimulating factor receptor (CSF1R) inhibitor within one week after induction of bAVM formation in mice not only depleted 93% of microglia within the brains but also inhibited the formation of the malformations [83]. Once the bAVMs were fully formed, the CSF1R inhibitor diminished the number of abnormal vessels and dwindled the microglia population inside the lesions themselves. This may propose the usefulness of CSF1R inhibitor in the prophylactic treatment of hemorrhage from brain bAVMs.

### 8.2. CCMs—The Story so Far

In an intraoperative electrocorticography study, the absence of spatially coincident bursts was correlated with a high density of HLA-DR activated microglia in patients with CCM [89]. This may indicate that M1 microglia are involved in the suppression of these discharge patterns in CCM patients, protecting them from epilepsy. However, in the patients with neurodevelopmental lesions, the high content of microglia was more often observed than in those with CCM, and so was Fe^3+^ staining, yet these findings could not be linked to any neocortical or hippocampal discharge patterns. Since microglia might safeguard neurons by storing excessive Fe^3+^ and by eliminating damaged cellular components, this would suggest that epileptogenesis in CCM is more complex than evaluated simply by histological studies. In a single case of operated cavernoma, Liu et al. found positive staining for the ribosomal protein phospho-S6 (pS6) in the reactive glia around the lesion itself [90]. pS6 is a downstream phosphorylated protein belonging to the mTOR pathway and is present within microglial cells in certain epilepsy-associated pathologies. It is noteworthy that epilepsy has been associated with a higher density of M1 activated microglia in patients with focal cortical dysplasia than in autopsy controls [91], although whether these immune cells are causative for epilepsy or simply grow in number as a response to it remains inconclusive.

To assess the degree of cerebral inflammation after acute hemorrhage via positron emission tomography (PET), Abid et al. utilized [^11^C]-(R)-PK11195 to highlight the perihematomal region [92]. [^11^C]-(R)-PK11195 attaches to translocator protein 18 kDa (TSPO), whose level is promptly increased in activated microglia. Two of their patients, one harboring a ruptured bAVM and the other a CCM, had biopsies revealing marked microglial activation within the perilesional tissue. Nevertheless, this finding does not necessarily indicate that microglia are involved in the growth of these vascular malformations as it has already been repeatedly shown that intraparenchymal hemorrhages are associated with a focal increase of the microglial cell population meant to mediate hematoma resolution [93,94,95,96,97]. Therefore, more research needs to be conducted to ultimately confirm the association between microglial cells and the development of intracranial malformations.

## 9. Limitations and Future Directions

Despite the growing interest in microglia and their involvement in cerebrovascular pathologies, the promising treatment methods presented in this review have yet to be translated into clinical trials on human patients. To the extent of our knowledge, only EPO has currently been used in human clinical trials for patients with aneurysmal SAH, with inconclusive results [98,99,100]. So far, no experimental studies of microglia-targeted therapies for CCMs have been performed. Moreover, some of the neuroinflammatory processes and mechanisms have not been irrefutably authenticated and remain conjectural. Table 1 summarizes the potential therapeutic agents targeting microglia and their signaling pathways in the pathologies discussed.

Modeling the complex cell-to-cell interaction of CNS development to study them under normal and pathological conditions involves the incorporation and co-culture of microglia with cerebral organoids [101]. Recently, several protocols for obtaining induced pluripotent stem cell (iPSC)-derived microglia have been successfully developed. This made it possible to study the long-term interaction between microglia and the cerebral tissue microevironment [101,102,103]. Currently, there are numerous models of organoids employed in the modeling of different types of pathologies [104]. Given the fact that vascularized brain organoids are in the early stages, the interaction between the vascular system and microglia, as well as its role in the emergence of intracranial aneurysms, AVMs, and CCMs, could be a future direction of research [105,106]. This will be particularly difficult in the case of a cerebral organoid, given that both cell types have a non-ectodermic origin. Lastly, the new organ/multi-organ-on-a-chip technologies, by incorporating biological and physical sensors, could provide essential information about these complex cell-cell interactions and may conclusively substantiate the exact mechanisms through which these enigmatic pathologies arise.

## 10. Conclusions

As of yet, the exact mechanisms through which microglia contribute to the generation and rupture of intracranial aneurysms and vascular malformations remain unclear and those which have been proposed are not definitively demonstrated, however great strides have been made in understanding microglial cell behavior in these pathologies. A sustained inflammation may lead to the evolution of these lesions, in conjunction with altered hemodynamics and stimulation via angiogenetic factors such as VEGF. TLR4 remains a highly feasible actor in both the mechanism of aneurysm development and an immunological cascade of SAH. Concerning bAVMs and CCMs, there is still much to be learned vis a vis the implications of microglia-derived chronic inflammation and whether this represents a valuable therapeutic target.

## Figures and Tables

**Figure 1 ijms-22-06141-f001:**
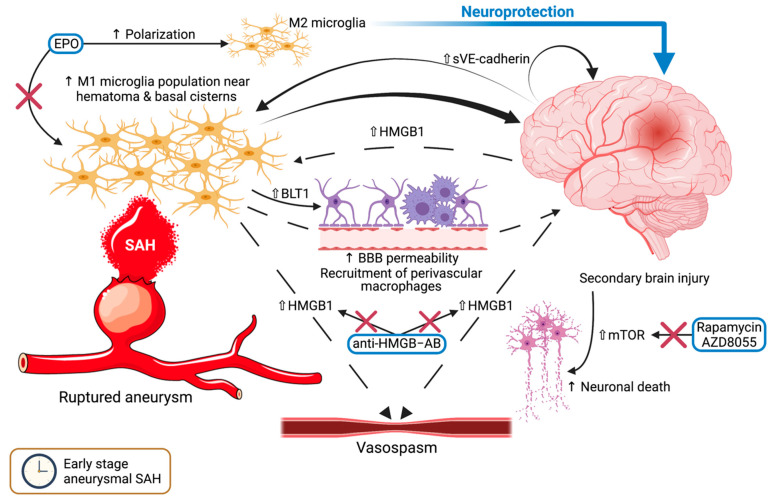
The mechanisms and pathways involving microglia in the early stage of subarachnoid hemorrhage. Following aneurysmal rupture, the blood in the subarachnoid and basal cisterns leads to an increased population of pro-inflammatory M1 microglia, which in turn will lead to secondary brain injury, heightened blood-brain barrier permeability, and recruitment of perivascular macrophages. Upregulated HMGB1 within the brain sustains this process and possibly leads to the occurrence of vasospasm, while anti-HMGB antibodies interrupt this cycle. VE-Cadherin maintains secondary brain injury and M1 microglia activation. Erythropoietin sways the polarization toward the M2 phenotype, resulting in a neuroprotective effect. Moreover, an upsurge of mTOR results in elevated neuronal death. Rapamycin and AZD8055 both inhibit the effects of mTOR. A transparent upward-pointing arrow denotes upregulation, whereas a thin upward-pointing arrow signifies an elevation of the mentioned process. Red crosses mark an inhibitory effect. Abbreviations (in alphabetical order): BBB, blood-brain barrier; BLT1, leukotriene B4 receptor 1; EPO, erythropoietin; HMGB1, high-mobility group box 1; HMGB1–AB, HMGB1-antibodies; mTOR, mammalian target of rapamycin; SAH, subarachnoid hemorrhage; sVE-Cadherin, soluble vascular endothelial cadherin.

**Figure 2 ijms-22-06141-f002:**
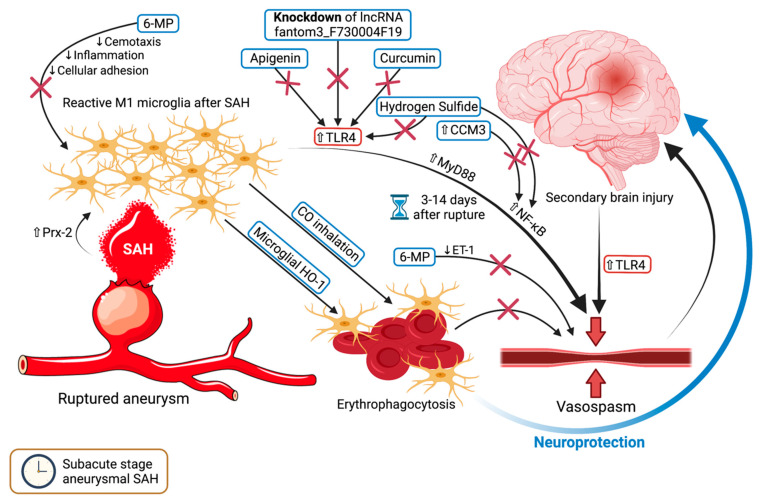
The mechanisms and pathways involving microglia in the subacute stage of subarachnoid hemorrhage (SAH), primarily vasospasm. The reactive M1 microglia that had been stimulated by SAH via Prx-2, in conjunction with the upregulation of TLR4, will result in an anarchic constriction of the cerebral vasculature, aggravating the preexisting brain injury. This occurs between days 3 and 14 after aneurysmal rupture. Certain agents have been shown to suppress the TLR4-dependent pathways by targeting TLR4, namely apigenin, curcumin, hydrogen sulfide, and the knockdown of the long non-coding RNA fantom3_F730004F19. Hydrogen sulfide and CCM3 upregulation also affect a downstream component of the pathophysiological cascade, namely NF-*κ*B. 6-mercaptopurine restricts vasospasm via endothelin-1 inhibition, while additionally reducing microglial-mediated, inflammation, chemotaxis, and cellular adhesion. CO inhalation and microglial HO-1 stimulate erythrophagocytosis, which in turn alleviates vasospasm and reduces brain damage. A transparent upward-pointing arrow denotes upregulation, whereas a thin downward pointing arrow signifies an inhibition of the mentioned process or protein. Red crosses mark an inhibitory effect. Abbreviations (in alphabetical order): 6-MP, 6-mercaptopurine; CO, carbon monoxide; CCM3, cerebral cavernous malformation 3 gene; ET-1, endothelin 1; HO-1, heme oxygenase isoform 1; lncRNA, long non-coding RNA; MyD88, myeloid differentiation primary response gene 88; NF-*κ*B, nuclear factor kappa-light-chain-enhancer of activated B cells; Prx-2, peroxiredoxin 2; SAH, subarachnoid hemorrhage; TLR4, toll-like receptor 4.

**Figure 3 ijms-22-06141-f003:**
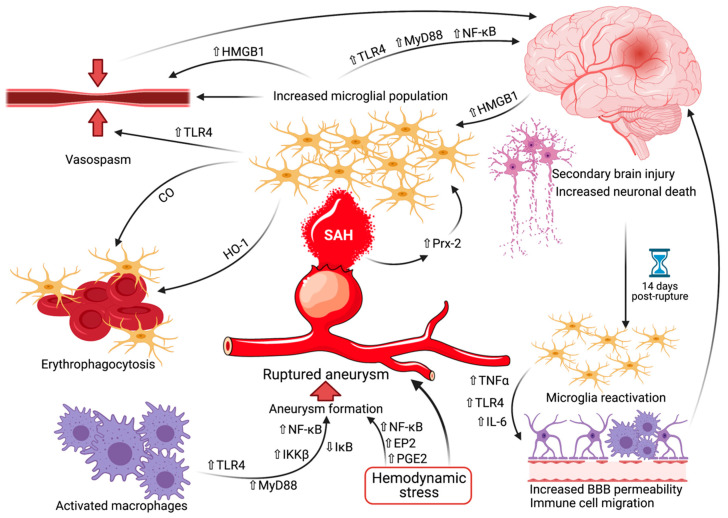
The pathophysiology of microglia in aneurysmal rupture. Activated macrophages in conjunction with hemodynamic stress lead to the formation of an intracranial aneurysm via pathways involving TLR4, MyD88, IKK*β*, I*κ*B NF-*κ*B, PGE2, and EP2. Hemodynamic stress will also eventually result in aneurysmal rupture. Aneurysmal SAH begets an increased reactive microglial population by way of Prx-2 upregulation. The microglia then generate secondary brain injury and subsequent increased neuronal death, partially also via vasospasm. At 14 days after hemorrhage, a reactivation of the microglia occurs, inducing increased BBB permeability and immune cell migration, further propagating the secondary brain injury. On the other hand, anti-inflammatory microglia are also responsible for the erythrophagocytosis at the site of SAH, diminishing its detrimental effect on the cerebral parenchyma. A transparent upward-pointing arrow denotes upregulation, whereas a transparent downward pointing arrow signifies downregulation. Abbreviations (in alphabetical order): BBB, blood-brain barrier; CO, carbon monoxide; EP2, prostaglandin E receptor 2; HMGB1, high-mobility group box 1; HO-1, heme oxygenase isoform 1; I*κ*B, inhibitor kappa B; IKK*β*, inhibitor kappa B kinase *β*; IL-6, interleukin 6; MyD88, myeloid differentiation primary response gene 88; NF-*κ*B, nuclear factor kappa-light-chain-enhancer of activated B cells; PGE2, prostaglandin E 2; Prx-2, peroxiredoxin 2; SAH, subarachnoid hemorrhage; TLR4, toll-like receptor 4; TNFα, tumor necrosis factor alpha.

**Table 1 ijms-22-06141-t001:** Potential microglial-targeted treatment agents in cerebrovascular pathologies.

Author, Year	Pathology	Agent/Drug	Target	Type of Study	Outcome
Wei et al., 2017 [46]	Aneurysm (acute stage SAH)	EPO	EPOR/JAK2/STAT3 pathway	in vitro (mouse brain SAH model)	Amelioration of brain injury.
Springborg et al., 2007 [98]	Aneurysm (subacute stage SAH)	EPO	NS	Double-blind randomized clinical trial	No conclusive evidence of beneficial effects from EPO.
Tseng et al., 2009 [99]	Aneurysm (subacute stage SAH)	EPO	NS	Double-blind randomized clinical trial	No differences in the incidence of vasospasm and adverse events; Patients receiving EPO had a decreased incidence of severe vasospasm.
Helbok et al., 2012 [100]	Aneurysm (vasospasm)	EPO	NS	Non-randomized clinical trial	Brain tissue oxygen tension improved; Brain metabolic parameters remained unchanged.
You et al., 2016 [48]	Aneurysm (acute stage SAH)	Rapamycin	mTOR	in vitro (rat brain SAH model)	Microglia polarization toward the M2 phenotype, early brain injury amelioration.
		AZD8055	mTOR	in vitro (rat brain SAH model)	Microglia polarization toward the M2 phenotype, early brain injury amelioration.
Wang et al., 2019 [58]	Aneurysm (acute stage SAH)	Anti-HMGB1–AB	HMGB1	in vitro and in vivo (rat brain SAH model)	Decreased cerebral vasoconstriction, improved cerebral blood flow, lessened brain edema, and microglial activation.
Heinz et al., 2021 [9]	Aneurysm (subacute stage SAH)	Perxidartinib	CSF1R	in vitro (mouse brain SAH model)	Reduced microglial accumulation and activation, diminished neuronal death.
Gao et al., 2019 [66]	Aneurysm (vasospasm)	Curcumin	TLR4	in vitro and in vivo (mouse brain SAH model)	Reduction of cerebral proinflammatory cytokines and edema; Promotion of M2 phenotype polarization; Diminished neurological deficit.
Duan et al., 2020 [67]	Aneurysm (vasospasm)	Hydrogen sulfide	TLR4/NF-*κ*B	in vitro and in vivo (rat brain SAH model)	Reduced the cognitive impairment; reduced the expression of TNF-α, TLR4, and NF-*κ*B
Zhang et al., 2015 [68]	Aneurysm (vasospasm)	Apigenin	TLR4/NF-*κ*B	in vitro and in vivo (rat brain SAH model)	Reduced neuronal apoptosis, inhibition of BBB disruption, and improved neurological results
Peng et al., 2017 [69]	Aneurysm (vasospasm)	Lentivirus vectors: lentivirus-50305 (KD1) and lentivirus-50307 (KD3)	lncRNA fantom3_F730004F19 and TLR4	in vitro (mouse brain SAH model)	Attenuated microglial inflammation.
Chang et al., 2011 [70]; Chang et al., 2010 [71]	Aneurysm (vasospasm)	6-Mercaptopurine	ET-1	in vitro (rat brain SAH model)	Inhibition of vasospasm, amelioration of microglial inflammation, and chemotaxis.
Schallner et al., 2015	Aneurysm (vasospasm)	CO inhalation	Microglial HO-1	in vitro and in vivo (mouse brain SAH model)	Amelioration of neuronal cell death, vasospasm, impaired cognitive function, and clearance of cerebral blood burden.
LeBlanc et al., 2016 [74]	Aneurysm (vasospasm)	Deferoxamine	Microglial HO-1	in vitro and in vivo (mouse brain SAH model)	Amelioration of neurological damage, cognitive outcome, and increased HO-1 expression in microglia, but no effect on vasospasm.
Ma et al., 2018 [83]	AVM	CSF1R inhibitor	CSF1R	in vitro (mouse brain AVM model)	Microglia depletion, AVM development inhibition.

Abbreviations (in alphabetical order): AVM, arteriovenous malformation; BBB, blood-brain barrier; CO, carbon monoxide; CSF1R, colony-stimulating factor 1 receptor; EPO, erythropoietin; EPOR, EPO receptor; ET-1, endothelin-1; HMGB1, high-mobility group box 1; HMGB1-AB, anti-HMGB–AB, anti-HMGB1 antibodies; HO-1, heme oxygenase isoform 1; I*κ*B, inhibitor kappa B; IKK*β*, inhibitor kappa B kinase *β*; IL-6, interleukin 6; JAK2, Janus kinase 2; lncRNA, long non-coding RNA; mTOR, mammalian target of rapamycin; MyD88, myeloid differentiation primary response gene 88; NF-*κ*B, nuclear factor kappa-light-chain-enhancer of activated B cells; NS, not specified; PGE2, prostaglandin E 2; Prx-2, peroxiredoxin 2; SAH, subarachnoid hemorrhage; STAT3, Signal transducer and activator of transcription 3; TLR4, toll-like receptor 4; TNFα, tumor necrosis factor alpha.

## Data Availability

Data sharing is not applicable to this article.

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
