# Peer review of "Involvement of Microglia in the Pathophysiology of Intracranial Aneurysms and Vascular Malformations—A Short Overview"

_ijms, 2021, doi:10.3390/ijms22116141_

Round 1
Reviewer 1 Report
This review article has summarized the current understanding of the contribution of microglia to neuroinflammation in intracranial aneurysms and vascular malformations. The article gives an interesting scientific perspective on this topic, focusing on the immune pathophysiology in different stages of intracranial aneurysms, including pre-rupture and post-rupture. Moreover, potential immunotherapies for different stages of intracranial aneurysms are discussed accordingly. The involvement of neuroinflammation in intracranial aneurysms and vascular malformations has long been discussed and similar reviews have been published, which makes this paper lacking in originality, however, this paper addressed further evidence supporting the importance of the function of microglia in intracranial aneurysms and vascular malformations.
There are several points that need to be further discussed or clarified in subsequent versions:
- Narrative reviews do not have a methods section but should include some information about applied methods at the end of the introduction.
- The context in section 4 could be improved by separating into paragraphs, such as M phenotypes and potential treatment respectively.
- The paragraphs in sections 5 and 6 are lack clear logic flow in the present version. Use subtitles might be one way of improving it. Or, for each stage of intracranial aneurysms, summarize the immune-inflammatory mechanisms in a figure will help readers to better follow the content.
- As mentioned above, the neuroinflammatory pathways or mechanisms underlying the diseases that have been discussed in the paper are not clearly demonstrated.
- Even though some abbreviations are commonly acknowledged, it is necessary to present their full names when firstly appeal in the paper, such as GFP (green fluorescent protein) on line 299.
- A figure legend for Figure 1 would make this figure easier to interpret.
Author Response
Cover letter for: ijms-1241175
Involvement of Microglia in the Pathophysiology of Intracranial Aneurysms and Vascular Malformations – A Short Overview
Esteemed editor and reviewers,
We, the authors, would like to express our gratitude for your kind remarks and suggestions for our manuscript. We hope that the changes made are to your expectations. If, however, you would like further changes, we are more than happy to comply. Below, you will find a point-by-point response to each suggestion, according to the reviewer.
Reviewer 1
This review article has summarized the current understanding of the contribution of microglia to neuroinflammation in intracranial aneurysms and vascular malformations. The article gives an interesting scientific perspective on this topic, focusing on the immune pathophysiology in different stages of intracranial aneurysms, including pre-rupture and post-rupture. Moreover, potential immunotherapies for different stages of intracranial aneurysms are discussed accordingly. The involvement of neuroinflammation in intracranial aneurysms and vascular malformations has long been discussed and similar reviews have been published, which makes this paper lacking in originality, however, this paper addressed further evidence supporting the importance of the function of microglia in intracranial aneurysms and vascular malformations.
There are several points that need to be further discussed or clarified in subsequent versions:
- Narrative reviews do not have a methods section but should include some information about applied methods at the end of the introduction.
Response: Thank you kindly for your remarks and suggestions. We have incorporated the Methods section within the Introduction as its last paragraph. As a result, the subsequent sections have been renumbered accordingly.
- The context in section 4 could be improved by separating into paragraphs, such as M phenotypes and potential treatment respectively.
Response: We have now divided section 4 into three paragraphs, the first discussing the two phenotypes in general, the second focusing on M1 polarization, and the third on M2 microglia and factors that stimulate this phenotype.
- The paragraphs in sections 5 and 6 are lack clear logic flow in the present version. Use subtitles might be one way of improving it. Or, for each stage of intracranial aneurysms, summarize the immune-inflammatory mechanisms in a figure will help readers to better follow the content.
Response: These sections have been divided into subsections focusing more clearly on the mechanisms, phenomena or pathways described. We have also divided the penultimate section (previously section 9) into two subsections, one for AVMs, the other for CCMs. Additionally, we have made 2 more figures, one for the early stage of SAH, and the other for vasospasm, with the appropriate legends. We hope you will find them suitable and informative for readers.
- As mentioned above, the neuroinflammatory pathways or mechanisms underlying the diseases that have been discussed in the paper are not clearly demonstrated.
Response: We have now specified this limitation in the conclusion, within a new section titled “Limitations and Future Directions,” and in several other places within the main sections. Please let us know whether further modifications are required regarding this point.
- Even though some abbreviations are commonly acknowledged, it is necessary to present their full names when firstly appeal in the paper, such as GFP (green fluorescent protein) on line 299.
Response: We apologize for this oversight, we have presented the full name of the abbreviation you mentioned, as well as for others (HLA-DR, EPOR, JAK2, STAT3, PET, and CO).
- A figure legend for Figure 1 would make this figure easier to interpret.
Response: Thank you for pointing this out, we have added a more detailed legend for figure 3 (formerly figure 1).
Again, we would like to thank the editor and the reviewers for their patience and suggestions.
Reviewer 2 Report
This article reviews the critical role of microglia in the pathophysiology of intracranial aneurysms and vascular malformations. The manuscript is well written and complete, including the most recent studies related to the field.
Minor Comments:
1-Authors could discuss how new technologies (iPSCs, Organoids, Organ-Chips) may help understand the involvement of microglia in these pathologies and their outcomes.
2-The authors should consider adding a table correlating treatment and targets. What is known so far, including data from clinical trials?
Author Response
Cover letter for: ijms-1241175
Involvement of Microglia in the Pathophysiology of Intracranial Aneurysms and Vascular Malformations – A Short Overview
Esteemed editor and reviewers,
We, the authors, would like to express our gratitude for your kind remarks and suggestions for our manuscript. We hope that the changes made are to your expectations. If, however, you would like further changes, we are more than happy to comply. Below, you will find a point-by-point response to each suggestion, according to the reviewer.
Reviewer 2:
This article reviews the critical role of microglia in the pathophysiology of intracranial aneurysms and vascular malformations. The manuscript is well written and complete, including the most recent studies related to the field.
Minor Comments:
1-Authors could discuss how new technologies (iPSCs, Organoids, Organ-Chips) may help understand the involvement of microglia in these pathologies and their outcomes.
Response: Thank you for your kind remarks and valuable insight. We have added a paragraph on these technologies in a new section, “Limitations and Future Directions.” However, as these technologies are new, we could not find articles specific for the pathologies discussed in our manuscript.
2-The authors should consider adding a table correlating treatment and targets. What is known so far, including data from clinical trials?
Response: We have also added a table regarding the treatments and targets presented in our manuscript, although, as far as we could find, only erythropoietin had been utilized in clinical trials.
Again, we would like to thank the editor and the reviewers for their patience and suggestions.
Round 2
Reviewer 1 Report
My comments have been appropriately addressed.